# Simultaneous Enhancement of Thermostability and Catalytic Activity of a Metagenome-Derived β-Glucosidase Using Directed Evolution for the Biosynthesis of Butyl Glucoside

**DOI:** 10.3390/ijms20246224

**Published:** 2019-12-10

**Authors:** Bangqiao Yin, Qinyan Hui, Muhammad Kashif, Ran Yu, Si Chen, Qian Ou, Bo Wu, Chengjian Jiang

**Affiliations:** 1State Key Laboratory for Conservation and Utilization of Subtropical Agro-bioresources, Guangxi Microorganism and Enzyme Research Center of Engineering Technology, College of Life Science and Technology, Guangxi University, Nanning 530004, China; yinbangqiao@163.com (B.Y.); hqyx114@163.com (Q.H.); Kashif_microbiologist@yahoo.com (M.K.); ouqian510@126.com (Q.O.);; 2Department of chemical and biological engineering, Guangxi Normal University for Nationalities, Chongzuo 532200, China

**Keywords:** metagenome-derived β-glucosidase, directed evolution, site-directed mutagenesis, thermostability, butyl glucoside

## Abstract

Butyl glucoside synthesis using bioenzymatic methods at high temperatures has gained increasing interest. Protein engineering using directed evolution of a metagenome-derived β-glucosidase of Bgl1D was performed to identify enzymes with improved activity and thermostability. An interesting mutant Bgl1D187 protein containing five amino acid substitutions (S28T, Y37H, D44E, R91G, and L115N), showed catalytic efficiency (*k_cat_*/*K*_m_ of 561.72 mM^−1^ s^−1^) toward *ρ*-nitrophenyl-β-d-glucopyranoside (*ρ*NPG) that increased by 23-fold, half-life of inactivation by 10-fold, and further retained transglycosidation activity at 50 °C as compared with the wild-type Bgl1D protein. Site-directed mutagenesis also revealed that Asp44 residue was essential to β-glucosidase activity of Bgl1D. This study improved our understanding of the key amino acids of the novel β-glucosidases and presented a raw material with enhanced catalytic activity and thermostability for the synthesis of butyl glucosides.

## 1. Introduction

β-1,4-Glucosidase is a pivotal rate-limiting enzyme of complex cellulose that breaks the β-1,4-glycosidic bond and generates diverse oligosaccharides, disaccharides, and alkyl β-D-glucosides [1]. In the past few years, industrial applications of β-glucosidase transglycosylation to produce alkyl glycosides have gained significant interest [2,3]. In particular, butyl glucoside is an excellent industrial cosolvent, which has favorable wetting, penetration, and decontamination properties [4]. The advantages of butyl glucoside biosynthesis, which utilizes simple procedures, moderate reaction conditions, and high product purity, have become increasingly prominent because of rapid developments in biotechnology [5]. However, butyl glucoside synthesis using bioenzymatic methods at high temperatures remains challenging [6].

Thermal stability is always an initial target for improving the attributes of industrial enzymes due to the multiple applications of thermostable enzymes in industrial processes [7,8], such as acceleration of the kinetic reaction, cost reduction of enzyme products, high stability, purification and packaging at room temperature, and improvement in substrate solubility [9]. Numerous studies have been conducted to understand the basic principles of protein stability, such as introducing disulfide bonds [10,11], salt bridges [12] and increasing intramolecular hydrophobic packing [13]. However, the application of these techniques has been restricted to enzymes with clearly known structures. Directed evolution has been approved as a powerful tool for studying and modifying the thermostability and catalytic activity of enzymes without extensive structural and functional information [2].

In the microbial community, the numbers of uncultured microorganisms (99% or more) greatly exceeds that of culturable microorganisms (1% or less) [14,15]. Using the metagenomics technique microbial genes from diverse environments without cultivable technology can be directly obtained [16,17]. Recently, β-glucosidase genes from environmental samples have been obtained via metagenomics technology [18,19,20]. The sources of uncultured microbes from which β-glucosidase genes have been isolated are mainly populated in the sea [20], compost [21], rumen and cecum [22,23], soil [19], spring water [24], and sewage [25]. A few glycosidase hydrolase family 1 (GH1) β-glucosidases from extreme thermal environments have shown the highest activity at 90 °C [24,26]. Most β-glucosidases have an optimum temperature in the range of 35–55 °C, such as some β-glucosidases from cow rumen metagenome (45 to 55 °C) [27], a β-glucosidase from buffalo rumen metagenome (37 °C) [28], and GH1 β-glucosidase from uncultured soil bacteria (55 °C) [29]. β-Glucosidase EglF2 from uncultured rumen of gayal has been detected at an optimal temperature of 55 °C and pH 6.2, however, its protein sequence displayed 93% homology to the known β-glucosidases in the GenBank database [22]. Equivalently, some GH3 β-glucosidases from rabbit cecum metagenome (40 to 55 °C) showed 50% to 70% similarities to known β-glucosidases of the GenBank [23]. Thus, among these enzymes, there is no β-glucosidase with an optimum temperature higher than 50 °C that shares less than 30% similarity to known β-glucosidases in the GenBank.

Interestingly, Bgl1D (GenBank accession number FJ686869), which was a novel β-glucosidase enzyme with no similarity with any known β-glucosidase according to the analysis data of NCBI BLAST, was purified and characterized [30]. Meanwhile, Bgl1D was a moonlight protein with lipolytic activity [31]. The utilization of this promising Bgl1D is limited due to its poor availability and low stability, although it is stable in a wide range of pHs. Protein engineering remains an effective method for modifying enzymes using the directed evolution method which enhances the novel Bgl1D applicability. Directed evolution of enzymes applies a method to the product biocatalysts for synthetically interesting transformations [32,33]. In this work, several interesting mutants with improved catalytic activities and thermostability were studied. In addition, site-directed mutagenesis was used to identify some key amino acids related to the catalytic activity and thermostability of β-1,4-glucosidase. Furthermore, the potential of thermostable enzymes for transglycosylation activity was also investigated.

## 2. Results

### 2.1. First and Second Round of Error Prone Polymerase Chain Reaction (epPCR) 

A 519 bps coding sequence from pETBlue-2-bgl1D carrying the *bgl1D* gene was subjected to the first round of epPCR. Through functional screening for higher catalytic activity, at a natural pH (7.0) with a temperature of 50 °C, approximately 5000 interesting colonies out of the 50,000 colonies showed black hydrolyzing zones on agar plates that contained ampicillin, esculin hydrate, and ferric ammonium citrate. These 5,000 colonies from the first round of epPCR library rapidly were tested for the activity of crude enzymes on 96-well plates. These 5000 colonies were cultured with 0.4 mM IPTG, induced, collected, and lysed by sonication. Their crude enzyme activities were detected by adding the substrate *ρ*NPG at 410 nm. After screening mutant β-glucosidases in the first round, three mutant β-glucosidases 2EP58 (Q25L/K117N/M148K), 6EP94 (S28P/I57K/E154G), and 20EP47 (F68L/I70M) were selected based on their higher activity (data not shown). 

The second-generation epPCR library was obtained when the interesting three mutants were used as templates. In the second-generation library, out of 50,000 positive colonies, the crude enzyme extracts of three unique interesting mutants, i.e., M2 (Q25L/S28T/L115Q/K117N/M148K), M6 (S28P/I57K/Y82S/W122G/E154G), and M20 (Y37H/D44E/F68L/I70M/R91G) showed that the hydrolytic activity was 2.3 to 3.6 times higher than that of Bgl1D, and 31% to 56 % residual activity was obtained after 2 h at 50 °C. However, the wild-type Bgl1D activity was undetectable at 50 °C (Appendix A). β-Glucosidases from mutants 2EP58, 6EP94, 20EP47, M2, M6, and M20 were overexpressed and purified in *Escherichia coli* BL21 (DE3) pLysS purified by His-tag chromatography, renamed as, Bgl1D58 (Q25L/K117N/M148K), Bgl1D94 (S28P/I57K/E154G), Bgl1D47 (F68L/I70M), Bgl1D2 (Q25L/S28T/L115Q/K117N/M148K), Bgl1D6 (S28P/I57K/Y82S/W122G/E154G), and Bgl1D20 (Y37H/D44E/F68L/I70M/R91G), respectively (Appendix A).

### 2.2. Directed Evolution

The purified β-glucosidases were incubated at 60 °C for various time intervals, and the remaining activity was determined, as shown in Table 1. Bgl1D2 (Q25L/S28T/L115Q/K117N/M148K) increased by roughly seven times in the half-life of thermal inactivation at 60 °C, which increased t_1/2_ of wild-type Bgl1D from 45 to 298 min. Bgl1D2 introduced two substitutions (S28T and L115Q) as compared with Bgl1D58 (Q25L, K117N, and M148K) whose half-life was almost the same as that of wild-type. Therefore, Ser28 and Leu115 were further studied. 

The hydrolytic kinetics of wild-type Bgl1D and mutants toward *ρ*NPG were determined at an optimal pH 10.0 and 30 °C (Appendix A), as shown in Table 2. The *k*_cat_, *K*_m_, *V*_max_, and *k*_cat_/*K*_m_ values of Bgl1D20 (Y37H/D44E/F68L/I70M/R91G) were 139.09 s^−1^, 0.47 mM, 153.70 U/mg, and 295.74 s^−1^ mM^−1^, respectively. Bgl1D20 introduced three substitutions (Y37H, D44E, and R91G) as compared with Bgl1D47 (F68L and I70M) whose kinetic constants were almost the same as the wild type. The result, characterized by kinetic properties toward *ρ*NPG, illustrated that Tyr37, Asp44, and Arg91 were the key amino acids which were further studied. 

### 2.3. Site Directed Mutagenesis 

The sites Ser28, Leu115, Tyr37, Asp44, and Arg91 were selected for further analysis. The site-directed mutagenesis assay of thermostability showed a three times increase of L115Q in the half-life of inactivation as compared with that of wild-type Bgl1D. We tried to mutate Leu115 to asparagine (N) as both asparagine (N) and glutamine (Q) were neutral amino acids. Interestingly, the site-directed mutagenesis assay of thermostability showed a four times increase of L115N in the half-life of inactivation as compared with that of wild-type Bgl1D. Meanwhile, S28T increased two times in the half-life of inactivation as compared with that of wild-type Bgl1D. The inactivation half-life of mutant S28T/L115N was 10 times greater than that of wild-type Bgl1D and proved to be more beneficial using site mutational analysis (Table 1). The kinetic parameters of these enzymes with site mutagenesis to *ρ*NPG are presented in Table 2. The *V*_max_ values of Bgl1D20 (F68L/I70M/Y37H/D44E/R91G) and Bgl1D47 (F68L/I70M) were 153.70 and 25.82 U/mg, respectively. There was no significant increase in enzyme activity as compared with wild type in site-directed mutagenesis to points Tyr37, Asp44, and Arg91, respectively (data not shown). Therefore, a combination of random mutant and site-directed mutagenesis approaches reveal that the best substitutions contained S28T, L115N, Y37H, D44E, and R91G, which had not only increased the t_1/2_ of Bgl1D, but also increased the *V*_max_ values of Bgl1D. This mutant was overexpressed and purified in *Escherichia coli* BL21 (DE3) pLysS purified by His-tag chromatography, renamed as, Bgl1D187 (S28T/Y37H/D44E/R91G/L115N). The experiments were also designed for these key amino acids, which were mutated to the amino acids of the opposite polarity, such as S28A, L115R, Y37G, D44G, and R91A. D44G enzyme did not show any activity toward *ρ*NPG (Table 2).

### 2.4. Comparison of Wild-Type Bgl1D and Bgl1D187 for pH and Temperature 

Bgl1D and Bgl1D187 (S28T/Y37H/D44E/R91G/L115N) exhibited high activity profiles at optimal pH values of 10.0 (Figure 1A). The pH stability curve for Bgl1D and Bgl1D187 showed that the enzymes were stable at a pH of 6.0 to 10.0 (Figure 1B). At an optimal pH 10.0, the mutant Bgl1D187 resulted in a shift in the optimal temperature to 50 °C (Figure 1C). The enzyme activity of Bgl1D187 still retained 95% of its activities after treatment at 60 °C for 1 h (Figure 1D). The t_1/2_ of Bgl1D187 was around 468 min, which was 10-fold higher than Bgl1D (Table 1).

### 2.5. Comparison of Wild-Type Bgl1D and Bgl1D187 for Kinetic Constants

The *K*_m_ values of Bgl1D187 (S28T/Y37H/D44E/R91G/L115N) (0.43 mM) were slightly lower than that of wild-type Bgl1D (0.54 mM), whereas *K*_m_ values of other enzymes were slightly higher (Table 2). Similarly, the *k*_cat_/*K*_m_ value of Bgl1D187 (561.72 mM^−1^ s^−1^) was significantly (*p* ≤ 0.5) higher than that of the wild type (25 mM^−1^ s^−1^). The *k*_cat_ value for Bgl1D187 enzyme was 242 s^−1^. These values were approximately 18-fold increased as compared with wild type (13 s^−1^). Interestingly, Bgl1D187 mutant showed a high *V*_max_ value (314 U/mg), which was 16-times higher than that of wild-type Bgl1D (20 U/mg). 

### 2.6. Comparison of Wild-Type Bgl1D and Bgl1D187 for Substrate Specificity

The specific activity towards various substrates of the purified protein of wild-type Bgl1D and mutant Bgl1D187 (S28T/Y37H/D44E/R91G/L115N) are presented in Table 3. Bgl1D belongs to the broad substrate-specific enzyme, considering that wild type and mutant enzymes also displayed broad substrate specificity toward aryl-glycosides, which contain β-Glc, β-Gal, and β-Xyl bond. The specific activity of mutant Bgl1D187 (335 U/mg) toward *ρ*NPG was increased 31-fold as compared with that of the wild type (11 U/mg). In addition, wild-type Bgl1D and mutant enzymes were specific to the substrates that contain Glc/Gal β-1, 4-Glc bond. Bgl1D6 (S28P/I57K/Y82S/W122G/E154G) totally lost the capability of hydrolyzed artificial and natural substrates that contained β-Gal bond, such as *ρ*NP-gal, *o*NP-gal, and lactose, while others were similar to the wild-type Bgl1D (Appendix A).

### 2.7. Comparison of Wild Type and Bgl1D187 for Organic Solvent Tolerance

The enzyme activity of wild type and mutants decreased with increasing concentrations of polar and nonpolar organic solvents (Appendix A). The enzyme activity of heat-resistant enzyme Bgl1D187 (S28T/Y37H/D44E/R91G/L115N) showed considerable resistance toward polar organic solvents, such as ethanol, isopropyl alcohol, butanol, benzene, toluene, and octanol (Figure 2). Meanwhile, the enzyme activity of Bgl1D187 was substantially simulated by different concentrations of butanol, which is an environmental friendly liquid biofuel produced by acetone-butanol-ethanol (ABE) fermentation [34]. Hence, butanol, a high-tolerant and promotive β-glucosidase Bgl1D187, could potentially be used for the production of butyl glucoside in transglycosylation and biobutanol in ABE fermentation.

### 2.8. Transglycosylation Production of Alcohols

Glucose of 20% (*w*/*v*) concentration and cellobiose were used as glycosyl donors because the enzyme activity of Bgl1D187 (S28T/Y37H/D44E/R91G/L115N) remained above 95% in the presence of 20% concentration of glucose (Appendix A). Methanol, ethanol, n-propanol, and butanol of 20% (*v*/*v*) concentrations were used as acceptor because the activity of heat-resistant enzyme Bgl1D187 was still very high (98%) at 20% concentration of alcohols (Figure 2). Only glucose could transfer the glycosyl moiety to butanol (Table 4). High-performance liquid chromatography (HPLC) results of the transglycosylation production confirmed the formation of butyl glucoside at 50 °C (Appendix A). The transglycosylation reaction showed three distinct peaks (3.112, 14.127, and 18.797 min) at 210 nm as compared with the control, in which two distinct peaks (3.078 and 14.071 min) were obtained. The third peak (18.797 min) in the transglycosylation reaction represented a compound with 5.5 milli absorbance unit (mAU), which corresponded to the product butyl glucoside. No transglycosylation production was detected in the presence of cellobiose.

## 3. Discussion

A mutant Bgl1D187 protein with decent properties in thermostability and catalytic activity toward *ρ*NPG, which contained five amino acid substitutions (S28T, Y37H, D44E, R91G, and L115N) was obtained using directed evolution and site-directed mutagenesis. The thermostability of enzymes was performed at different temperatures. As compared with Bgl1D58 (Q25L/K117N/M148K), the t_1/2_ of Bgl1D2 (Q25L/S28T/L115Q/K117N/M148K) increased seven-fold. Hence, newly introduced substitutions (Ser28Thr and Leu115Gln) could be related to the thermal inactivation of half-lives of Bgl1D. The site-directed mutagenesis shows that S28T significantly increased the half-life of inactivation of Bgl1D (Table 1). Therefore, threonine, which replaced serine at position 28, contributed to the increase in the half-life of inactivation. Moreover, Bgl1D6 (S28P/I57K/Y82S/W122G/E154G), which harbored Ser28Pro, remained constant in the t_1/2_. In addition, sequence statistics indicated that threonine was the best residue that interacted with water molecular-surrounding protein structures and this interaction was one of the predominant structural factors responsible for protein thermostability [35,36]. The change in Leu115 to hydrophilic amino acid asparagine contributed to the increase in the enzyme thermostability (Table 1). Site-directed mutagenesis showed that L115N significantly increased the half-life of inactivation of Bgl1D. The Leu444 substitution of a β-glucosidase from *Thermobifida fusca* enhanced thermostability, which attributed to the number of putative hydrogen bonds that were increased at this position [37]. Correspondingly, the 115 position changes of Bgl1D could introduce more hydrogen bonds, causing the interactions among surface residues, which were strengthened; these interactions were significant for the maintenance of the skeleton rigidity and protein stability [38]. The thermostability of Bgl1D187 (S28T/Y37H/D44E/R91G/L115N) was higher than the activity of β-glucosidase BglC from *Thermobifida fusca*, which were obtained by family shuffling [37].

The *V*_max_ of Bgl1D20 (Y37H/D44E/F68L/I70M/R91G) significantly increased to 295.74 U/mg as compared with Bgl1D47 (F68L/I70M). Hence, newly introduced substitutions (Y37H/D44E/R91G) could be related to the catalytic activity in Bgl1D. Site-directed mutagenesis assay showed no obvious enhancement in the catalytic activity when mutate residues Tyr37, Asp44, and Arg91 were used. Therefore, a mutant Bgl1D187 was obtained for the next study, which had five substitutions (S28T, Y37H, D44E, R91G, and L115N).

The result of site-directed mutagenesis of D44G indicated that the substitution of Asp44 with glycine led to the complete inactivation of the enzyme, whereas β-glucosidase activity of D44E remained the same, as Asp44 and Glu44 shared the same carboxyl group in the side chain (Table 2). The amino acid alignments of Bgl1D, with the most similair β-glucosidases in the PDB database, were performed using a SWISS-MODEL server. Five glycosidases from the GH3 (PDB 3U48 and 5K6M), GH9 (PDB 1JS4 and 3X17), and GH116 (PDB 5BX5) families that shared the highest similarity to Bgl1D were selected for the alignment (Appendix A). Blast results showed that Bgl1D shared approximately 16.57%, 27.49%, 24.56%, 20.78%, and 26.67% identities with the amino acids of 3U48, 5K6M [39], 1JS4 [40], 3X17 [41], and 5BX5 [42], respectively. A comparison of the amino acid of Bgl1D with these β-glucosidases demonstrated that the Asp44 of Bgl1D was similar to 3U48 (GH3 family) and 1JS4 (GH9 family) whose residue Asp was generally used as conserved catalytic nucleophile base or catalytic proton donor according to the information of β-glucosidase (EC 3.2.1.21) from the CAZy database (http://www.cazy.org) (Figure 3) [43]. Thus, Asp44 residue, as a conserved catalytic nucleophile and base, was necessary for Bgl1D activity. 

The enzyme activity of Y82S showed a greater increase in *K*_m_ (2.13 mM) with the substrate *ρ*NPG as compared with Bgl1D (0.54 mM), suggesting that replaced aromatic amino acids Tyr82 by Ser82 decreased the affinity between the substrate (*ρ*NPG) and the enzyme (Table 2). The site-directed mutagenesis of Y82S showed its β-glucosidase activity toward *ρ*NPG decreased to 9.53 U/mg, while that of wild type was 20 U/mg. Data statistics showed that Tyr occurred at the highest frequencies in the active site architecture of β-glucosidases [44].

The newly introduced substitutions, Tyr82Ser and Trp122Leu, from the second round of epPCR loss the ability to catalyze β-galacosidic bond (Appendix A). In addition, the site-directed mutagenesis showed that Y82S did not affect β-galacosidic bond degradation, but W122G displayed zero specificity activity to substrate of *ρ*NP-gal, *o*NPG, and lactose (Appendix A). Moreover, C4 hydroxyl moiety of glucose was hydrogen-bonded by Trp, and Trp residue was strictly conserved in β-galactosidase [45]. Therefore, Trp122 residue could be a binding site of β-galacosidic substrate in Bgl1D. In order to clearly understand the relationship between protein structure and function of Bgl1D, the three-dimensional (3D) structure of protein was determined using X-ray diffraction, but no Bgl1D crystal was found. 

In this work, the mutant Bgl1D187 (S28T/Y37H/D44E/R91G/L115N) showed strong resistance toward polar and nonpolar organic solvents as compared with wild type (Figure 2). Consistent with this finding, a thermostable mutant of fructose bisphosphate aldolase showed resistance to polar and nonpolar organic solvents [46]. However, thermostable mutants of *Bacillus liceniforms* lipase only have shown improved resistance to nonpolar organic solvents [47]. The enzyme activity of the heat-resistant enzyme, Bgl1D187, showed considerablely improved resistance toward polar organic solvents, such as ethanol, isopropyl alcohol, butanol, benzene, toluene, and octanol (Figure 2). This was similar to the activity of BGL from *Bacillus subtilis* which showed a slight increase under 10% and 30% ethanol [48] and that of β-glucosidase from *Thermoascus aurantiacus*, which showed 54-fold higher activity toward *ρ*NPG with the addition of 20% octanol [49].

Similar to our findings, Watt et al. [50] showed that β-glucosidase from *Agrobacterium tumefaciens* was highly tolerant to butanol and only butyl glucoside was found in transglycosylation assay. Until the performance of the current procedures, no report was conducted about alkyl glycoside synthesis above 50 °C. Transglycosylation products were both formed by a β-glucosidase from *Trichoderma harzianum* and a β-glucosidase from *Hanseniaspora thailandica* BC9 under 30 °C [51,52]. β-Glucosidase efficiently synthesizes cellotriose and cellotetraose at 40 °C [53], however, in our work, the most thermostable mutant Bgl1D187 (S28T/Y37H/D44E/R91G/L115N) retained transglycosidation activity at 50 °C (Table 4). Some single mutations (D44G and W122G) were deleterious. The directed evolution of enzymes as catalysts improve from these nonadditive cooperative mutational effects in protein engineering [54]. Currently, ongoing studies are expected to obtain crystals of Bgl1D and mutant enzyme with cellobiose, *ρ*NPG, glucose, and thiocellobiose. Future research should focus on improving the products of butyl glucoside of Bgl1D187 by more rounds of directed evolutions.

## 4. Materials and Methods

### 4.1. Construction and Screening of Mutant Libraries

Error prone polymerase chain reaction (epPCR) conditions for the generation of mutagenic *bgl1D* gene were implemented according to Cadwell and Joyce’s description with some modifications [55]. EpPCR was performed in 100 μL of the reaction solution containing 1 ng/μL pETBlue-2 as template from previous study with 3 mM MgCl_2_ (Sigma-Aldrich, Inc., Darmstadt, Germany) and 0.3 mM MnCl_2_ (Sigma-Aldrich, Inc.) [30]. To this solution, 0.1 μM each of forward and reverse primers with *Eco*RI and *Hin*dIII restriction enzyme sites (underlined) (5′-ACGAATTCATGAGCTTATTTTTTACTTTTAATCCAT-3′/5′-TCAAGCTTTTAGTTAACTCCTATATTAACGATATTA-3′) (New England Biolabs, Ipswich, MA, USA) was added. The PCR products were digested at restriction enzyme sites, ligated into plasmid pET-32a(+) (Novagen, Darmstadt, Germany), introduced into the host bacteria *E. coli* BL21 (DE3) pLysS (Novagen, and spread on LB agar plate supplemented with 0.2% esculin (*w*/*v*), 0.05% ferric ammonium citrate (*w*/*v*), and 100 μg/mL ampicillin (Sigma-Aldrich, Inc., Darmstadt, Germany). The colonies with black phenotype were transferred into 10 mL bottles with 0.4 mM isopropyl β-d-thiogalactopyranoside (IPTG) (Sigma-Aldrich, Inc.). After collecting bacteria, cells lysed by sonication in 1 mL 20 mM Tris-HCl (pH 8.0) (Sigma-Aldrich, Inc.), were centrifuged at 12,000 rpm for 10 min. The crude enzyme activities of the mutants were detected at 410 nm in an Epoch microplate spectrophotometer (BioTek Instruments, Inc., Winooski, VT, USA). From the first round of epPCR, three mutant vectors that embraced the beneficial mutations were used as the templates for the second round of epPCR, respectively. The mutants having high β-glucosidase activities from the second round of epPCR were further studied. The interesting mutations from two rounds of epPCR were sequenced at Sangon Biotech Co., Ltd., Shanghai, China.

### 4.2. Construction of Site-Directed Mutants

According to the comparison of two rounds of epPCR, the site-directed mutagenesis was performed for newly introduced amino acids. The protocol of Fast Mutagenesis System Kit (TransGen Biotech Co. Ltd., Beijing, China) was used for the construction of mutants for point mutation. Appendix A displays the primer sequences, which contain the proper base substitutions. The successful site mutations were verified by sequencing at Sangon Biotech Co., Ltd., Shanghai, China.

### 4.3. Overexpression and Purification of the Proteins

The plasmid pET-32a(+) and *E. coli* BL21 (DE3) plysS were used as expression vector and strain and were cultured in LB medium that contains ampicillin (100 μg/mL) and carbenicillin (50 μg/mL) (Sigma-Aldrich, Inc.). The culture was induced with 0.4 mM of IPTG when the OD_600_ of the bacterial culture reached 0.6. Protein purification were performed according to Jiang et al. as previously description [30].

### 4.4. Determination of Enzyme Activity of the Proteins

The β-glucosidase activity was measured using *ρ*NPG (Merck KGaA, Darmstadt, Germany) as the substrate. The enzyme reaction mixture was comprised of 14 μL of 25 mM *ρ*NPG, 10 μL of appropriately diluted enzyme, and 116 μL of glycine-NaOH buffer incubated at an optimal temperature for 0.5 h (Sigma-Aldrich, Inc.). The reaction was stopped by adding 70 μL of Na_2_CO_3_. The released amount of *ρ*NPG was observed at 410 nm in Epoch microplate spectrophotometer (BioTek Instruments, Inc., Winooski, VT, USA). One unit (U) of β-glucosidase activity was defined as the amount of enzyme that releases 1 μmol *ρ*NP per minute under the corresponding reaction conditions.

### 4.5. Determination of the Effect of pH and Temperature on the Enzyme Activity

The effect of pH and temperature on enzyme activity was determined at a pH of 3.0 to 12.0 and temperatures from 20 to 80 °C, respectively. The highest enzyme activity was measured at an optimal pH and temperature, which is defined as 100%. The effect of pH and temperature on enzyme stability was tested by incubating the enzyme at a pH of 3.0 to 12.0 and 20 to 80 °C for 24 and 1 h without the substrate. The half-life of inactivation of enzyme was tested at 60 °C. 

### 4.6. Kinetic Parameters of the Enzyme Proteins

The kinetic parameters of β-glucosidase toward *ρ*NPG (1 to 50 mM) were measured at an optimal pH 10.0 and optimal temperatures of 30 and 50 °C. The *K*_m_ and *V*_max_ values of enzyme proteins were calculated with GraphPad Prism 5.0 software (GraphPad Software, La Jolla, CA, USA).

### 4.7. Substrate Specificity Assays of Enzyme Proteins

Specificity of the mutants was measured by incubating the purified protein in glycine-NaOH (pH 10.0) that contains 1 mM of aryl-glycosides and saccharides (BioTek Instruments, Inc.) at 30 and 50 °C for 0.5 h. Under the standard assay condition, the released *ρ*NP and *o*NP were determined. Moreover, the enzyme activity on lactose was determined by measuring the amount of reduced sugars with the dinitrosalicylic acid method.

### 4.8. Transglycosylation Activity of the Most Thermostable Enzyme

The transglycosylation activity of interesting mutant was performed in 5 mL of reaction solution, which contains 40 μg/mL (final concentration) of heat-resistant enzyme, 1 g of glucose, 3.8 mL of glycine-NaOH buffer (pH 10.0), and 20% (*v*/*v*) methanol, ethanol, n-propanol, and butanol, at 50 °C for 12 h with 200 r/min (Sigma-Aldrich, Inc.). The reaction solution without enzyme was used as a control. Water HPLC system equipped with a reverse-phase NH_2_ column (Alltima, 250 mm × 4.6 mm, 5 μm), a Model e2695 separation module system, and a Model 2998 photodiode array detector (Waters Corporation, Milford, CT, USA) was used to detect the transglycosylation product. The sample was injected and eluted with isocratic elution of 83% acetonitrile in water at a flow rate of 1 mL/min, at 25 °C and 210 nm.

## 5. Conclusions

Directed evolution, in combination with site mutagenesis, for protein engineering of the metagenome-derived β-glucosidase Bgl1D revealed a series of mutants with enhanced activity and thermostability. Ser28, Tyr37, Asp44, Arg91, and Leu115 were identified as the key amino acids in Bgl1D, and mutant Bgl1D187 simultaneously showed improved catalytic activity and thermostability. Moreover, the most thermostable Bgl1D187 exhibited an efficient transglycosylation capacity which allowed the synthesis of butyl glucoside at high temperatures using glucose as donor.

## Figures and Tables

**Figure 1 ijms-20-06224-f001:**
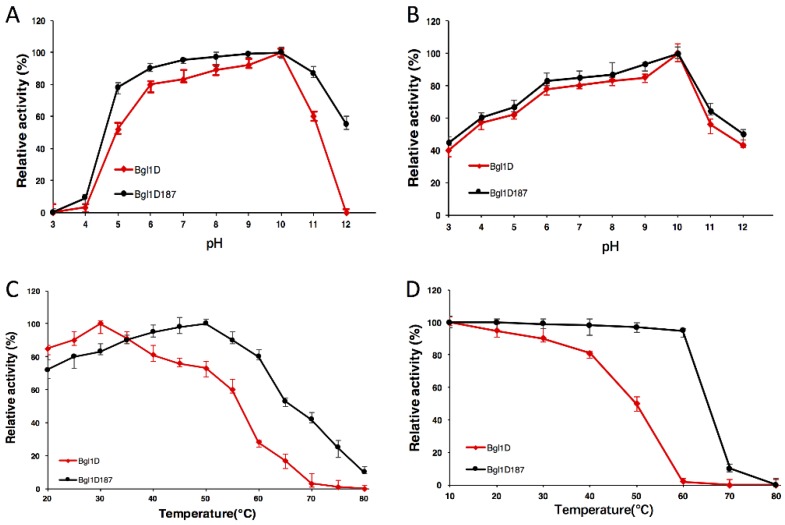
Enzymatic properties of β-glucosidases of wild type and Bgl1D187 using *ρ*NPG as the substrate. (**A**) Effects of pH on enzyme activity. The enzyme activities were measured at 37 °C and a pH of 3.0 to 12.0 in 0.1 M of buffer (pH 3.0 to 8.0, Na_2_HPO_4_-citric acid buffer; pH 8.6 to 10.6, glycine-NaOH buffer; and pH 10.9 to 12.0, Na_2_HPO_4_-NaOH buffer). (**B**) Effect of pH on enzyme stability. The enzyme was mixed with 0.1 M of buffers at pH 3.0 to 12.0 and incubated at 4 °C for 24 h. (**C**) Effects of temperature on the enzyme activity. The enzyme activities were measured at 20 to 80 °C and pH of 10.0. (**D**) Effects of temperature on the enzyme stability. The enzymes were incubated at 20 to 80 °C for 1 h.

**Figure 2 ijms-20-06224-f002:**
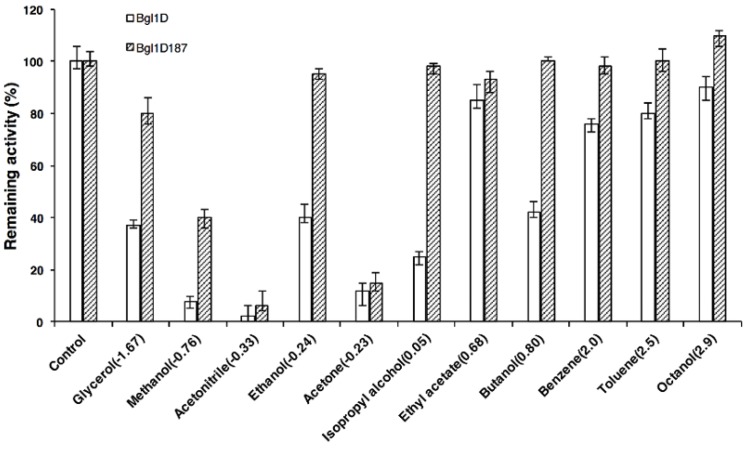
Effect of 30% concentration of organic solvents on the Bgl1D and Bgl1D187 enzymes. The residual activity after incubation was measured using *ρ*NPG as substrate. The number in the parenthesis indicated the polarity value of relevant solvents.

**Figure 3 ijms-20-06224-f003:**
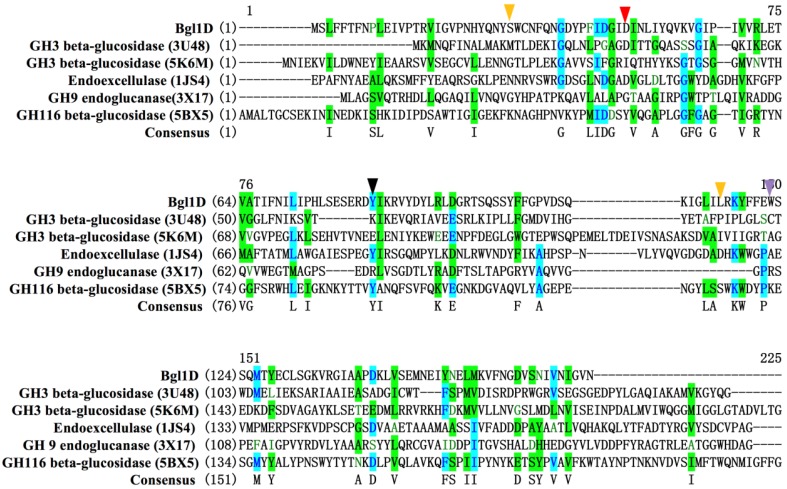
Sequence alignment of Bgl1D with other known β-glucosidases from GH3, GH9, and GH116 glycosidase sequences. The identical residues were highlighted in blue. The weakly similar residues were highlighted in green. The conserved catalytic Asp44 residue is indicated with a red triangle. The residue Tyr82 and residue Trp122 are indicated by black and purple triangles, respectively. Two residues, Ser28 and Leu115, related to thermostability are indicated by orange triangles. The sequences from top to bottom are GH3 β-glucosidase from compost metagenome (PDB ID 3U48), GH3 β-glucosidase from cow rumen metagenome (PDB ID 5K6M), GH9 exocellulase from *Thermomonospora fusca* (PDB ID 1JS4), GH9 endoglucanase from uncultured bacterium (PDB ID 3X17), and GH116 β-glucosidase from *Thermoanaerobacterium xylanolyticum* (PDB ID 5BX5).

**Table 1 ijms-20-06224-t001:** Half-life thermal inactivation for the wild type Bgl1D and mutants at 60 °C.

Enzyme	Mutations	t_1/2_ (min)	Relative t_1/2_
Bgl1D	None	45	1
Bgl1D58	Q25L/K117N/M148K	46.32	1
Bgl1D 94	S28P/I57K/E154G	45.71	1
Bgl1D 47	F68L/I70M	45.98	1
Bgl1D2	Q25L/S28T/L115Q/K117N/M148K	298.02	7
Bgl1D6	S28P/I57K/Y82S/W122G/E154G	45.48	1
Bgl1D20	Y37H/D44E/F68L/I70M/R91G	46.34	1
Bgl1D28	S28T	105.9	2
Bgl1D115q	L115Q	136.74	3
Bgl1D115	L115N	183.75	4
Bgl1D28115	S28T/L115N	447.85	10
Bgl1D187	S28T/Y37H/D44E/R91G/L115N	468.21	10

**Table 2 ijms-20-06224-t002:** Kinetic constants of Bgl1D and mutant enzymes toward *ρ*NPG.

Enzyme	Mutations	*k*_cat_ (s^−1^)	*K*_m_(mM)	*V*_max_ (U/mg)	*k_cat_*/*K*_m_ (s^−1^ mM^−1^)
Bgl1D	None	13.40 ± 0.19	0.54 ± 0.03	20.10 ± 0.06	24.81 ± 0.22
Bgl1D58	Q25L/K117N/M148K	23.84 ± 1.92	1.25 ± 0.12	21.73 ± 2.46	14.45 ± 0.95
Bgl1D94	S28P/I57K/E154G	48.14 ± 2.34	1.17 ± 0.05	52.94 ± 4.23	41.13 ±2.67
Bgl1D47	F68L/I70M	29.38 ± 3.53	0.53 ± 0.08	25.82 ± 6.36	55.43 ± 3.22
Bgl1D2	Q25L/S28T/L115Q/K117N/M148K	31.22 ± 1.04	1.74 ± 0.14	37.46 ± 0.65	17.94 ± 0.03
Bgl1D6	S28P/I57K/Y82S/W122G/E154G	55.80 ± 6.05	1.67 ± 0.12	42.50 ± 3.80	33.41 ± 2.60
Bgl1D20	Y37H/D44E/F68L/I70M/R91G	139.09 ± 8.05	0.47 ± 0.02	153.70 ± 2.50	295.74 ± 1.31
Bgl1D44	D44G	0	0	0	-
Bgl1D82	Y82S	5.34 ± 0.08	2.13 ± 0.0.73	9.53 ± 0.11	2.51 ± 0.02
Bgl1D28115	S28T/L115N	16.53 ± 2.86	1.01 ± 0.45	26.33 ± 3.21	16.37 ± 3.14
Bgl1D187	S28T/Y37H/D44E/R91G/L115N	241.54 ± 10.07	0.43 ± 0.04	314.30 ± 2.25	561.72 ± 4.50

**Table 3 ijms-20-06224-t003:** Substrate specificity of Bgl1D, Bgl1D6, and Bgl1D187 enzymes.

Substrate	Linkage of Glycosyl Group	Specific Activity (U/mg)	
		Bgl1D	Bgl1D187
*Aryl-glycosides*			
ρNP-β-d-glucopyranoside	βGlc	10.8	335.88 ± 0.21
ρNP-β-d-galactopyranoside	βGal	7.34 ± 0.07	16.72 ± 0.82
oNP-β-d-galactopyranoside	βGal	4.86 ± 0.54	7.93 ± 1.72
ρNP-α-d-glucopyranoside	αGlc	ND ^a^	ND ^a^
ρNP-N-acety-β-d-glucosaminide	βGlc	0.85 ± 0.83	2.87 ± 0.57
ρNP-β-d-xylopyranoside	βXyl	5.70 ± 0.22	ND ^a^
Salicin	βGlc	2.84 ± 0.76	38.99 ± 1.55
*Saccharides*			
Sophorose	Glcβ(1,2)Glc	ND ^a^	ND ^a^
Cellobiose	Glcβ(1,4)Glc	1.40 ± 0.04	181.57 ± 0.10
Lactose	Galβ(1,4)Glc	2.38 ± 0.48	31.5 ± 0.10
Trehalose	Glcα(1,1)Glc	ND ^a^	ND ^a^
Maltose	Glcα(1,4)Glc	ND ^a^	ND ^a^
Isomaltose	Glcα(1,6)Glc	ND ^a^	ND ^a^
Mannose	Glcα(1,4)Glc	ND ^a^	ND ^a^
Sucrose	Glcα(1,2)Fru	ND ^a^	ND ^a^
Xylan	βXyl	8.64 ± 0.64	10.66 ± 0.55
CMC	βGlc	ND ^a^	19.19 ± 0.36
Soluble starch	αGlc	ND ^a^	ND ^a^
Starch from wheat	αGlc	ND ^a^	ND ^a^
4-Methylumbelliferyl-β-d-glucopyranoside	βGlc	^−b^	^−b^

ND ^a^, not detected and ^−b^, fluorescence can be detected.

**Table 4 ijms-20-06224-t004:** Transglycosylation activity results of Bgl1D187.

Acceptors	Donors	
	Glucose	Cellobiose
Methanol	ND ^a^	ND ^a^
Ethanol	ND ^a^	ND ^a^
*n*-Propanol	ND ^a^	ND ^a^
Butanol	^−b^	ND ^a^

ND ^a^, not product detected and ^−b^, product detected.

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
