# Peer review of "Simultaneous Enhancement of Thermostability and Catalytic Activity of a Metagenome-Derived β-Glucosidase Using Directed Evolution for the Biosynthesis of Butyl Glucoside"

_ijms, 2019, doi:10.3390/ijms20246224_

Round 1

Reviewer 1 Report

The authors present a well-performed study of directed evolution of a metagenome derived -glucosidase as a catalyst for the production of butyl glucoside. Two rounds of epPCR were carried out, in addition to site directed mutagenesis for the formation of single mutants. One of the best mutants was characterized by 5 point mutations. The goals of the study were nicely reached, higher activity and moderately enhanced thermostability being reported.

Publication is recommended following minor revision: The authors present some interesting evidence that certain single mutations are deleterious. This touches on the traditional question of additivity versus non-additivity of mutations and sets of mutations as revealed by complete deconvolution of multi-mutational variants. It may help the reader to assess the whole data if the authors point this question out and cite the minireview of Reetz, Angew. Chem. Int. Ed. 2013, 52, 2658-2666, or the most recent review on methods in directed evolution by G. Qu, et al, Angew. Chem. Int. Ed. 2019; DOI: 10.1002/anie.201901491

Author Response

Response to Reviewer 1 Comments

Point 1: The authors present a well-performed study of directed evolution of a metagenome derived -glucosidase as a catalyst for the production of butyl glucoside. Two rounds of epPCR were carried out, in addition to site directed mutagenesis for the formation of single mutants. One of the best mutants was characterized by 5 point mutations. The goals of the study were nicely reached, higher activity and moderately enhanced thermostability being reported.

Response 1: We thank you for affirming the significance of our study.

Point 2: Publication is recommended following minor revision: The authors present some interesting evidence that certain single mutations are deleterious. This touches on the traditional question of additivity versus non-additivity of mutations and sets of mutations as revealed by complete deconvolution of multi-mutational variants. It may help the reader to assess the whole data if the authors point this question out and cite the minireview of Reetz, Angew. Chem. Int. Ed. 2013, 52, 2658-2666, or the most recent review on methods in directed evolution by G. Qu, et al, Angew. Chem. Int. Ed. 2019; DOI: 10.1002/anie.201901491

Response 2: Thank you for this valuable suggestion. Accordingly, we have cited the above papers as the following described:

1)         “Directed evolution of enzymes apply a method to product biocatalysts for synthetically interesting transformations [33].” (New Version Line: 73-75).

2)         “Some single mutations (D44G and W122G) are deleterious. The directed evolution of enzymes as catalysts improve from these non-additive cooperative mutational effects in protein engineering [54].” (New Version Line: 331-333).

Qu, G.; Li, A.; Sun, Z.; Acevedo-Rocha, C. G.; Reetz, M. T., The crucial role of methodology development in directed evolution of selective enzymes. Angew Chem Int Ed Engl 2019. Reetz, M. T., The importance of additive and non-additive mutational effects in protein engineering. Angew Chem Int Ed Engl 2013, 52, (10), 2658-66.

Reviewer 2 Report

The authors of the present manuscript report the mutational studies of a novel metagenomic-derived beta-glucosidase, in order to obtain a more robust biocatalyst for application in trans glycosylation reactions. The experiments have been conducted appropriately, however, the manuscript presents severe weaknesses as far as the presentation and argumentation related to the results are concerned.

Overall, the language throughout the manuscript needs substantial improvement.

The weakest part is the “Discussion”:

First, it is not very easy for the reader to follow. It would be helpful if the point mutations were indicated in parenthesis next to each mutant name. In addition, there are several mistakes: eg in page 7, line 208, the reported number is wrong. Also, lines 213-216: there is no D44E mutant in Table 2.

Figure 3 has at least two sequences that are identical (pdb codes 5BX5, 5BVU). The authors should indicate the %homology of the included enzymes with their beta-glycosidase and explain why they chose these ones for the alignment.

Reference 38 does not include any “LIDG” motif and residue D is not an active site residue as suggested by the authors. Please correct accordingly.

PAGE 7, LINES 233-235: there is no Y28S mutant in table 2.

Page 8, line 240: table S2 does not contain the related information.

Other corrections:

Page 2, line 47: What do the authors mean by the term “sophisticated”? Please rewrite the corresponding phrase

Page 2, line 56: remove ref. no 24 after “soil”

Page 2, lines 70-73: “The use of these..properties”. These two phrases should be rewritten. Explain “its stability was in a wide range of pHs”.

Page 2, line 81: define “natural pH”

Page 2, paragraph 2.1, 1st paragraph: it is not clear how 5000 out of 50000 colonies were selected first, and then three out of these 5000. Please make it more clear for the reader.

Page 3, line 112: why was leucine 115 mutated to asparagine instead of glutamine, as was done in the 1stround of screening of improved mutants.

Table 2 should be reformulated so the correlation between mutant name and mutations is more evident.

Page 4, line 116-117: the reported numbers and mutations are wrong.

Figure 2: please define what is the number in the parenthesis next to the solvent name.
